# Optimal Hyperdimensional Representation for Learning and Cognitive Computation

## Abstract

Hyperdimensional Computing (HDC), as a novel neurally-inspired computing methodology, uses lightweight and high-dimensional operations to realize major brain functionalities. Recent HDC works mainly focus on two aspects: brain-like learning and cognitive computation. However, it lacks differentiation between these functions and their requirements for HDC algorithms. We address this gap by proposing an adaptable hyperdimensional kernel-based encoding method. We explore how encoding settings impact HDC performance for both tasks, highlighting the distinction between learning patterns and retrieving information. We provide detailed guidance on kernel design, optimizing data points for accurate decoding or correlated learning. Experimental results with our proposed encoder significantly boost image classification accuracy from 65% to 95% by considering pixel correlations and increase decoding accuracy from 85% to 100% by maximizing pixel vector separation. Factorization tasks are shown to require highly exclusive representation to enable accurate convergence.

## 1 Introduction

The human brain remains the most sophisticated processing component that has ever existed, albeit after more than decades of advancement in computer science. The ever-growing research in biological vision, cognitive psychology, and neuroscience has given rise to many concepts that have led to prolific advancement in artificial intelligent accomplishing cognitive tasks (Lindsay, 2020; Indiveri & Horiuchi, 2011; Mitrokhin et al., 2020). Particularly, brain-inspired machine learning methods have shown promising leads in realizing crucial brain functionalities thanks to the advancement in theoretical neuroscience. Among these, a more recent and actively-studied direction is Hyperdimensional Computing (HDC), a computing framework that mimics the brain at abstract and functionality level (Kanerva, 2009). HDC uses high-dimensional representations that are holographic, i.e., the information encoded is evenly distributed across all dimensions. More importantly, HDC enjoys the advantages of structured and symbolic vector representations through a well-defined set of algebraic operations in the high-dimensional space, i.e., *Hyperspace*. The vector representation within the hyperspace is usually referred to as *Hypervectors*. Notable models in the HDC family are Tensor Product Representations, Holographic Reduced Representations (Tay et al., 2019), Multiply-Add-Permute (Kleyko et al., 2021), Binary Spatter Codes (Kleyko et al., 2016), and Sparse Binary Distributed (Rachkovskiy et al., 2005).

Several recent efforts focus on mapping HDC to various learning and cognitive tasks. For example, HDC is leveraged for several machine learning tasks, including classification (Najafabadi et al., 2016), clustering (Imani et al., 2020), regression (Hernández-Cano et al., 2021), fault detection (Poduval et al., 2021a; 2022a), and face detection (Imani et al., 2022; Poduval et al., 2021b). Similarly, HDC shows advances in reasoning and cognitive operations (Poduval et al., 2022b). With a suitable encoder in each task that maps data into hyperspace, the learning and cognitive computations are carried out using basic HDC mathematics almost linearly, thereby leading to optimal results.

Even though encoders are crucial for representing knowledge appropriately, researchers nowadays can only empirically select the HDC encoding method for each task; because it is unclear how different HDC encoder designs should interact with input information and patterns. More specifically, *how should one select a suitable encoding for HDC? Does HDC encoding depend on the desired task? Can a single encoding support all tasks based on HDC?*

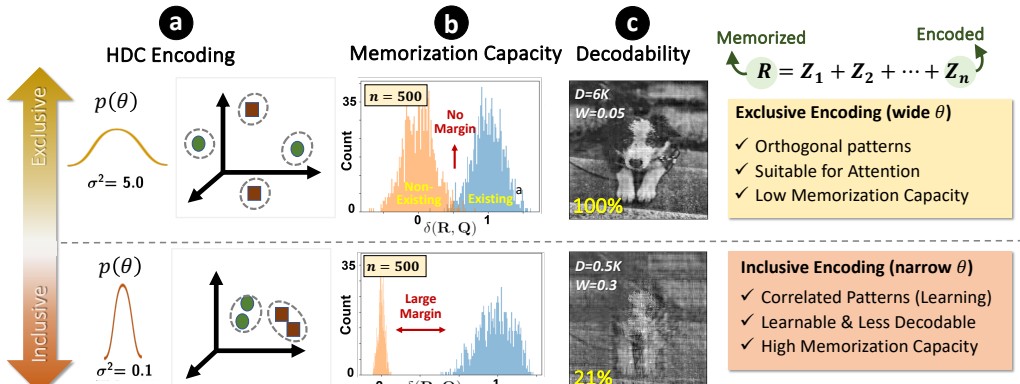

Figure 1: Two directions in HDC encoding designs: the correlative one is suitable for learning and the exclusive one is suitable for cognition.

We first observe that computation in hyperspace may require different representations depending on the nature of the task, e.g., pattern extraction for learning or accurate information retrieval for performing cognitive reasoning. For learning tasks, a typical example is an image classification or object detection task, in which the model identifies the object and predicts its category. The model focuses on extracting high-level features from images. Therefore, the encoded hypervectors are not required to store unhelpful or redundant information for learning purposes. In contrast, cognitive tasks focus more on reasoning, answering questions about relationships between objects, and making traceable and justifiable decisions. These tasks often require preserving most information of original data to ensure accurate information retrieval, in other words, decoding. In this work, we try to enable HDC learning and cognitive computation using the same encoding flow, and the main contributions of the paper are:

- We propose a universal hyperdimensional encoding method that can be easily adapted toward high-quality learning or accurate information retrieval. On the contrary, in existing HDC encoder designs, we observe a large number of empirically selected encoding algorithms that achieve good results on various tasks. Yet, they are generally not compatible with each other. Our proposed HDC encoder is more flexible for different applications and saves the design cost.

- We provide the first rigorous theoretic analysis of the fundamental requirement of two tasks of very different natures: learning and cognition. We define a separation metric that represents how encoded data points are separated or correlated in the hyperspace. Our analysis suggests that the learning task requires data to be encoded in a correlated way while decoding in cognitive tasks requires encoding different data points separately.

- We carry out extensive explorations on our universal encoder, in which we adjust several knobs according to the derived separation metric. We verify our theoretic analysis by observing how the quality of learning and decoding changes when the data is encoded with or without correlations. When the vectors representing each pixel are made orthogonal by tuning down the scale $w$, the decoding procedure produced near-accurate results. On the other hand, the learning procedure produced accurate results only in a high $w$ regime.

In our experiments, we find that the image classification accuracy significantly increases from 65% to 95% (with separation changing from 0.3 to 1.2) when we take into consideration the correlation between pixels. On the other hand, the decoding accuracy increases from 85% to 100% when we maximize the separation of vectors representing the pixels (with corresponding separation changing from 1.0 to 4.0). In practice, we find that the decoding task requires a higher separation of about 2 to 3 for accurate results because it is highly sensitive to noise while the learning task requires only a low value of 0.8 to 1.2 for the best accuracy. For factorization of hypvervectors into hyperspace, the hypervectors require a highly exclusive representation to recover the correct factors, and any correlations can induce errors in the corresponding factors.

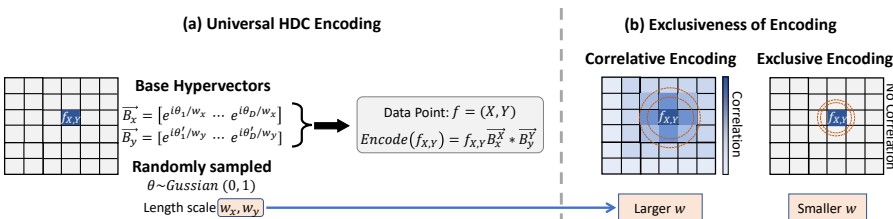

Figure 2: Our universal encoding can easily adjust the exclusiveness of the encoding using the knobs in the Gaussian kernel.

## 2 HYPERDIMENSIONAL COMPUTING: AN OVERVIEW

HDC uses large dimensional hypervectors to represent information within such a hyperspace using nearly orthogonal hypervectors (Kanerva, 1998). Information is combined through hypervectors using well-defined vector space operations, e.g., *Bundling (+)* and *Binding (∗)*. Bundling uses element-wise addition to represent sets, and binding expresses conjunctive association with element-wise multiplication. The hypervectors are holographic and (pseudo)random with i.i.d. components, allowing a holistic representation so that information is spread across all of the components.

In recent years, HDC has been employed in a range of applications, such as classification (Kanerva, 2009), activity recognition (Kim et al., 2018), biomedical signal processing (Moin et al., 2021), multimodal sensor fusion (Räsänen & Saarinen, 2015), security (Thapa et al., 2021; Zhang et al., 2021) and distributed sensors (Kleyko et al., 2018). A key HDC advantage lies in the capability of training in one or few shots, where object categories are learned from a few examples without many iterations and has achieved SOTA compared to support vector machines (SVMs), gradient boosting, and convolutional neural networks (CNNs) (Rahimi et al., 2018; Mitrokhin et al., 2019), as well as lower execution energy on embedded processors (Montagna et al., 2018).

In these successful HDC applications, HDC encoding is essential to the quality of computing. The encoding determines (1) the distance metric for encoded data points and (2) the level of correlation or exclusiveness preserved in hyperspace. In this work, we introduce the HDC encoder which can tune the level of inclusiveness in the encoding to hyperspace, and study the decodability and learning capabilities of the resulting model. In Fig. 10, we categorize HDC applications into learning and cognition, then we shed light on two corresponding directions in HDC encoding.

- **Learning:** aims to capture the general pattern of data. The appropriate encoding should abstract common information by keeping the similarity of neighboring data points in hyperspace, and we refer to this as the *Correlative Encoding*, since it should correlate hypervectors depending on the underlying correlations in feature space, thus classifying data that are not linearly separable and avoid overfitting with a smooth boundary. The correlative encoder also theoretically increases the memorization and learning capacity of hypervectors.

- **Cognition:** aims to represent structural data using neural patterns, which accordingly enables brain-like analysis and information extraction, requiring an exclusive representation of data in hyperspace. We call this the *Exclusive Encoding*, ensuring accurate knowledge extraction such that memorized information can be used as prior information for various cognitive computation tasks. The encoded information needs to be accurately decoded back to the original space to answer cognitive questions.

As shown in Fig. 10, learning and cognitive computation have different requirements for HDC encoders. For learning, the encoding is inclusive and correlative, preserving the similarity of nearby data. In contrast, encoding is exclusive for cognitive computation, where data points are mapped to orthogonal space and distinct.

## 3 UNIVERSAL NEURAL ENCODING

We now discuss the correlative encoder that is most commonly used to encode spatio-temporal data and sequences in a correlative manner, with the example of image encoding. The encoder works by introducing a kernel-generated HDC basis vector for each location in the image [Fig. 2(a)]. These

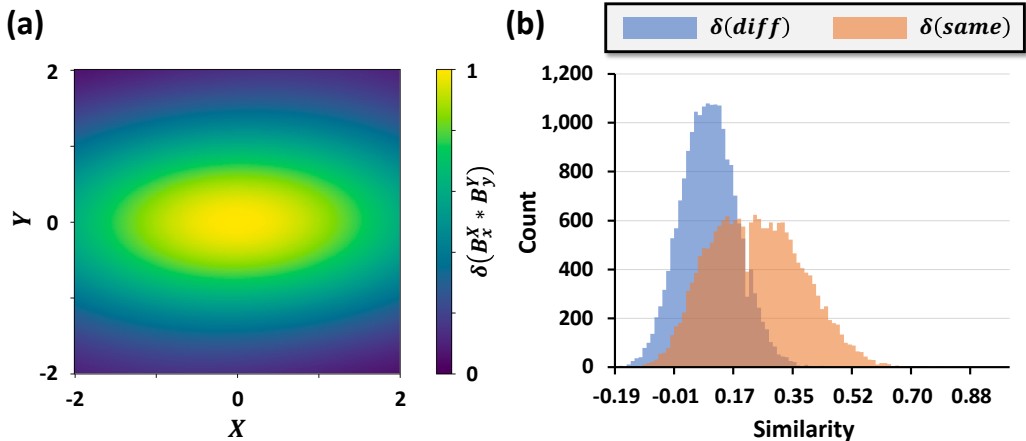

Figure 3: (a) The 2D kernel approximated by HDC similarity with length scales $w_x = 2$ and $w_y = 1$, whose shape is wider in the $x$ direction as compared to the $y$ direction (b) The distribution of similarity for images within the same class and across different classes.

position basis vectors are correlated with each other based on the distance between the location, and the correlation decays with distance based on two parameters, $w_x$ and $w_y$, which are the length scales over which the kernel decays in the $x$ and $y$ direction, respectively, and are tunable parameters. If $w_x$ and $w_y$ are significant, then the kernel (and thus, the HDC basis vectors) remains correlated over a considerable distance in the image, making the resulting encoding more correlative and better capture global features necessary for learning [Fig. 2(b)]. A smaller value of $w_x$ and $w_y$ makes the basis vectors more exclusive since the position basis vectors are now independent and uncorrelated. As a result, each data point will end up being uncorrelated in HDC space, allowing the *decoding* of data from the hypervector. As a result, there is a natural tradeoff between the correlativeness and decodability of the HD encoding vector, which is controlled primarily by $w_x$ and $w_y$.

In this work, our goal is to analyze the correlative nature and the decodability of the HD encoding process and characterize its dependence on the scale of the kernel ($w_x$ and $w_y$). First, we formally define the universal hyperdimensional encoding process that prepares encoded data for learning and cognition. Suppose that the input of the encoder is a 2D image $f$, with $f_{X,Y}$ representing the pixel at position $(X, Y)$. We randomly generate two basis hypervectors $\vec{B}_x$ and $\vec{B}_y$ as $\vec{B}_x = e^{i\vec{\theta}_x/w_x}$ and $\vec{B}_y = e^{i\vec{\theta}_y/w_y}$, where $\theta \in \{\mathcal{N}(0,1)\}^D$ is sampled from the $D-$dimensional normal distribution, and $w_i$ are the length scales which determine the correlation of the position vectors. To represent a certain position $(X_1, Y_1)$ in the image, we define the corresponding hypervector $B_x^{X_1} * B_y^{Y_1}$, where $*$ is the elementwise product between the two hypervectors. The resulting basis vectors are correlated through the Gaussian kernel, with $\delta(B_x^{X_1}, B_x^{X_2}) \overset{D\to\infty}{\approx} k(\frac{X_1 - X_2}{w_x})$ (where $k(r) = e^{-\frac{r^2}{2}}$ is the standard Gaussian kernel) and similarly $\delta(B_y^{Y_1}, B_y^{Y_2}) \overset{D\to\infty}{\approx} k(\frac{Y_1 - Y_2}{w_y})$. The kernel's exact form is unimportant; If $\theta$ is sampled from a general distribution $p(\theta)$, the corresponding kernel is $k(x) = \int d\theta p(\theta) e^{i\theta x}$(Bochner, 1946). The kernel ensures that the location hypervectors are correlated between nearby pixels and thus helps maintain spatial information during the encoding. Finally, the image $f$ is encoded to its corresponding hypervector $\vec{\mathcal{V}}_f$ as:

$$\vec{\mathcal{V}}_f = \sum_{X,Y} f_{X,Y} B_x^X * B_y^Y. \tag{1}$$

In Fig. 3, we show the kernel function in 2D with $w_x = 2$ and $w_y = 1$. The kernel spreads further in the $x$ direction, being closer to 1 for $|x| \lesssim 1$, since the length scale is twice in the $x$ direction as compared to the $y$ direction. In Fig. 3(b), we show the similarity distribution using a small synthetic image classification dataset, where the similarities between images of the same class has the average value significantly larger than 0, signifying that they are closely related. On the other hand, for images across different classes, the average similarity is generally lower than in the other case, showing a clear separation between the two distributions.

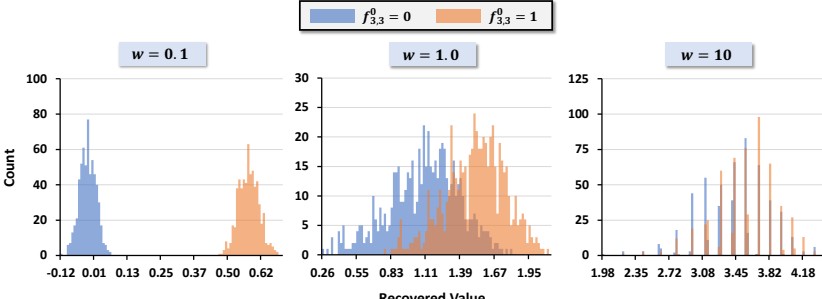

Figure 4: Distribution of the decoded values for different values of $w$. As $w$ increases, the separations between the two distributions decrease.

## 4 INFORMATION RETRIEVAL FOR COGNITION

In this section, we focus on a crucial step when leveraging HDC for cognitive tasks: accurately retrieving information from hypervectors. In our example about image datasets, the information retrieval process aims at decoding the original pixel values $f_{X,Y}$ from the corresponding hypervector $\vec{\mathcal{V}}_f$. Our method is inspired by the prior work (Poduval et al., 2022b) that uses HDC algorithms for knowledge extraction and information compression. The decoding is an iterative process, where the estimates of $f_{X,Y}$ are used to approximate the noise and refine the estimates in the next cycle. In practice, unless the kernel encoding is highly exclusive, the decoding process is highly error prone for continuous features. Therefore, the features need to be quantized, with the spacing between the quantized values in proportion to the noise. For simplicity purposes, we will quantize the pixel values to binary values of 0 or 1, which will also enable us to understand the decoding noise analytically. We stress that the binarization procedure does not restrict our experiments or understanding and is done only for simplicity, and can be easily extended to a general quantization.

The decoding process is described as follows. First, we make an initial estimation of the feature value, defined as $f_{X,Y}^0$, and calculated as $f_{X,Y}^0 = \text{Binarize}[\delta(B_x^X * B_y^Y, \vec{\mathcal{V}}_f)]$, where the binarization function binarizes the value to 0 if it is less than mean of $\delta(B_x^X * B_y^Y, \vec{\mathcal{V}}_f)$, and 1 otherwise. Using this, we construct the first estimate of the encoded vector $\vec{\mathcal{V}}_{f^0}$ as $\vec{\mathcal{V}}_{f^0} = \sum_{X,Y} f_{X,Y}^0 B_x^X * B_y^Y$. The first estimate of the encoded hypervector can predict the noise in the encoding and then iteratively cancel the noise. The corresponding recursive equation to refine the decoded values is given by $f_{X,Y}^n = \text{Binarize}[\delta(B_x^X * B_y^Y, \vec{\mathcal{V}}_f - \vec{\mathcal{V}}_{f^{n-1}}) + f_{X,Y}^{n-1}]$, which is repeated till convergence.

During information retrieval, the length scale $w$ plays a crucial role in considering the correlations of nearby pixels. A large $w$ should be used only when correlations are exceptionally high in neighboring pixels, otherwise, a small $w$ should be used. To explore its effect on the information retrieval process, we rewrite the initial estimate for pixel $f_{X,Y}$ as the following:

$$f_{X,Y}^0 = f_{X,Y} + \underbrace{\sum_{X' \neq X, Y' \neq Y} f_{X',Y'} \delta(B_x^X * B_y^Y, B_x^{X'} * B_y^{Y'})}_{Noise \approx N(\mu, \sigma)}, \tag{2}$$

where $f_{x,y}$ is considered the retrieved information, and the rest of the terms resulting from the kernel are considered as noise. However, in the presence of correlations, the "noise" can better recover the information. Assuming the worst case of uncorrelated neighouring pixels, the noise is approximated by the Central Limit Theorem as a normal $N(\mu, \sigma)$ distribution with $\mu = \frac{1}{2} \sum_{X' \neq X, Y' \neq Y} k\left(\frac{X-X'}{w_x}\right) k\left(\frac{Y-Y'}{w_y}\right)$ and $\sigma^2 = \frac{1}{4} \sum_{X' \neq X, Y' \neq Y} \left[k\left(\frac{X-X'}{w_x}\right) k\left(\frac{Y-Y'}{w_y}\right)\right]^2$.

To better understand the meaning behind these equations, we use a tiny image with size $5 \times 5$ as an example. We consider the cases where the center pixel is either 0 or 1, and randomly generate the rest of pixels. In Fig. 4, we plot the distribution of the center pixel $f_{3,3}^0$ based on equation 2. We assume $w_x = w_y = w$ and vary it between 0.1 and 10. Our expectation is that the initial estimate $f_{3,3}^0$ have

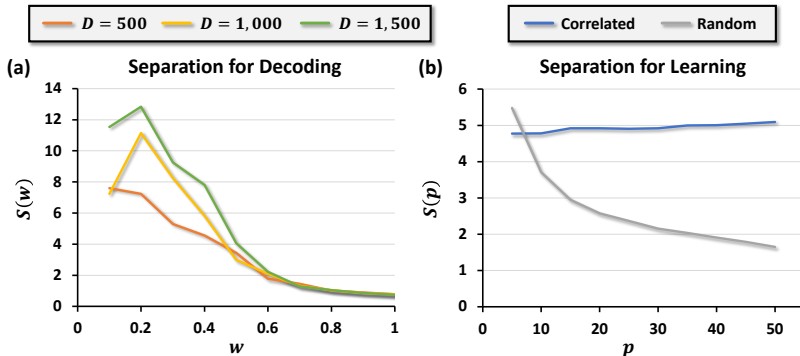

Figure 5: (a) Separation $s(w)$ for different values of dimension $D$ in the case of decoding and (b) $s(p)$ Separation as a function of number of learning data points for the case of memorization.

a distribution which is well-separated between cases when $f_{3,3} = 0$ and $f_{3,3} = 1$. Fig. 4 shows that the separation mainly depends on the length scale $w$. When $w$ is very small, then the distribution is well-separated because it is assumed that there is no correlation. However, the distributions get much closer to each other when $w$ becomes larger. To have minimum overlap, we need the sum of the standard deviation to be much lower than the difference between their means. Based on this intuition, we define a separation between two distributions as

$$s = \frac{\mu_2 - \mu_1}{\sigma_1 + \sigma_2}, \tag{3}$$

with $\mu_i, \sigma_i$ properties of the signal and noise distribution during the decoding process for $i = 2, 1$ respectively. They inherently depend on the parameters of the encoding $D$ and $w$, and for optimizing our design we study the variation of separation as a function of the encoding parameters. With that view, plot the separation $s(w)$ as a function of $w$ in Fig. 5(a) and observe that as $w$ increases, the separation $s$ decreases. In practice, the calculation of the separation metric depends on the data set of interest, and the distribution followed by the feature values. If some prior distribution can be assumed about the features of the dataset, then the separation metric can be analytically calculated to understand the optimum kernel width.

## 5 HDC Memorization and Pattern Extraction for Learning

In this section, we introduce the Hyperdimensional learning algorithm and provide insights on how to adjust the HDC encoder for learning tasks. In HDC classification, we can correctly identify a class if the noise distribution is well separated from the signal distribution. A heuristic measure for the separation is if the sum of the standard deviation of both distributions (which can be visualized as the width of the distributions) is smaller than the difference of the corresponding means of the distributions.

In order to analyze the capacity of class hypervectors in HDC classification tasks, we can make an assumption on the prior distributions of the data points. As an example, we consider a image dataset with two classes, with the corresponding class hypervectors being $\vec{\mathcal{C}}_1$ and $\vec{\mathcal{C}}_2$. The class hypervector is constructed by bundling vectors of the same class as $\vec{\mathcal{C}}_i = \sum_{j \in \text{Class}i} \vec{\mathcal{H}}_j$. The dimensionality of these hypervectors remains as $D$. To take into account the correlation, we consider the similarity between hypervectors of class 1 to follow a distribution $D_1(\mu_1, \sigma_1)$, and similarly a distribution $D_2(\mu_2, \sigma_2)$ for class 2. The similarity between the two classes is defined as $D_{12}(\mu_{12}, \sigma_{12})$. We use $D_i(\mu_i, \sigma_i)$ to refer to generic distributions characterized by the mean $\mu_i$ and standard deviation $\sigma_i$. We also make an assumption that $\mu_{12} < \mu_1, \mu_2$; this is because we expect the similarity between two images belonging to the same class is higher than those belonging to different classes.

During the training, $p$ training images are stored in class hypervectors, i.e., $\vec{\mathcal{C}}_1$ and $\vec{\mathcal{C}}_2$, according to their labels. For example, if an encoded query $\vec{\mathcal{Q}}$ belongs to class 1, the similarity to $\vec{\mathcal{C}}_1$ follows a normal distribution given by $N(p\mu_1, \sqrt{p}\sigma_1)$, because of the central limit theorem with large sample number $p$. Recall that the class hypervector $\vec{\mathcal{C}}_1$ is constructed by bundling together $p$ image

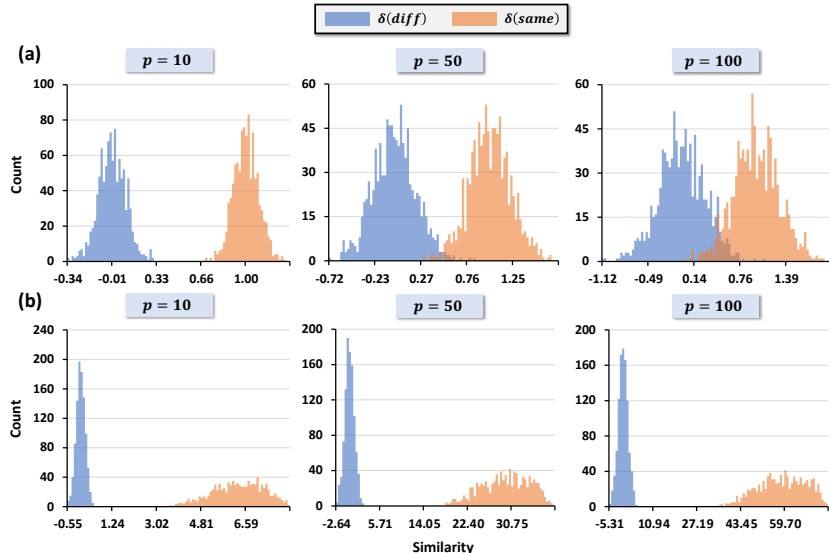

Figure 6: The signal and noise distribution for (a) exclusive encoding suitable for cognition and (b) correlated encoding for different values of $p$, suitable for learning.

hypervectors belonging to the training set. Similarly, the similarity of $\vec{\mathcal{Q}}$ with $\vec{\mathcal{C}_2}$ follows the cross-class normal distribution $N(p\mu_{12}, \sqrt{p}\mu_{12})$. Thus, we can measure the noise separation of these two distributions as follows:

$$s = \frac{p\mu_1 - p\mu_{12}}{\sqrt{p}\sigma_1 + \sqrt{p}\sigma_{12}} = \sqrt{p}\frac{\mu_1 - \mu_{12}}{\sigma_1 + \sigma_{12}}. \tag{4}$$

If the average points are located far away, the distributions have minimum overlap. In other words, the distributions are well separated. This simple calculation shows that the separation value increases with the number of training samples as $s \propto \sqrt{p}$. Thus, adding more samples enables us to better memorize the data if the correlations are well-preserved. We plot the separation for the case of learning in Fig. 5(b), which shows that in the case of random encoding, the separation decreases to 0 as the number of data points increases. However, in the case of correlated encoding, the separation remains reasonably large. In Fig. 6(a), we show the signal and noise distribution for a toy-correlated dataset with random encoding, and in (b) we show the signal and noise distribution with correlated encoding for different values of $p$.

## 6 MEMORISING ASSOCIATIONS

HDC can represent associations in a robust and holographic manner through the process of binding. If $\vec{\mathcal{A}}$ and $\vec{\mathcal{B}}$ are two nearly orthogonal bipolar vectors, then $\vec{\mathcal{S}} = \vec{\mathcal{A}} * \vec{\mathcal{B}}$ will be nearly orthogonal to both $\vec{\mathcal{A}}$ and $\vec{\mathcal{B}}$ due to the inherent randomness, which can be used to represent structures like key-value pairs(Poduval et al., 2022b), sequences (Zou et al., 2022; Poduval et al., 2021c), and data with multiple properties(Frady et al., 2020). For example, suppose we want to memorize an object, the location it was observed, the time it was observed, and its size in an exclusive manner, we can represent each feature with a hypervector and bind them together. The objects can be sampled from a codebook $\mathcal{O} = \{\vec{\mathcal{O}}_1, .., \vec{\mathcal{O}}_n\}$. Each hypervector could represent an object like a ball, cat, dog, apple, etc. The position, time and size components are, however, continuous valued. The corresponding values can be encoded individually using the kernel hypervectors as described in the previous sections. To encode the position, time and size $(x, t, s)$, we assign randomly sampled base vectors $(B_X, B_T, B_S)$ to each component, where $\vec{B}_i = e^{i\vec{\theta}_i/w_i}$ with $\vec{\theta} \in \{\mathcal{N}(0, 1)\}^D$. The encoding of $(x, t, s)$ is then $(\vec{B}_X^x, \vec{B}_T^t, \vec{B}_S^s)$. Finally, the association between the object $O$ and its features $(x, t, s)$ is memorised by the hypervector $\vec{\mathcal{H}} = \vec{\mathcal{O}}_i * \vec{B}_X^x * \vec{B}_T^t * \vec{B}_S^s$.

Given a hypervector $\vec{\mathcal{H}}$, decomposing it into the factorization $\vec{\mathcal{H}} = \vec{\mathcal{O}}_i * \vec{B}_X^x * \vec{B}_T^t * \vec{B}_S^s$ is a non-trivial problem. The state of the art approach is the resonator network (Frady et al., 2020), which is a recurrent network that iteratively calculates a guess for the hypervectors, and uses the guesses to

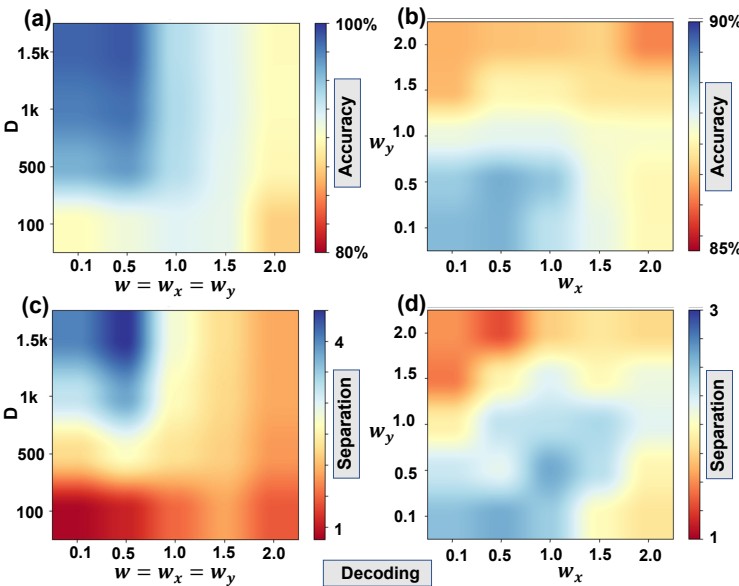

Figure 7: The decoding accuracy as (a) function of $D$ and $w$ (b) function of $w_x$ and $w_y$; the decoding separation as (c) function of $D$ and $w$ (d) function of $w_x$ and $w_y$

cancel the noise to make subsequent guesses more accurate. Suppose that the possible position, time and size values are from a list $\{x_1, .., x_n\}, \{t_1, .., t_n\}$ and $\{s_1, .., s_n\}$ respectively. Then, having calculated a set of factor guesses at the $(n-1)^{th}$ iteration labelled by $\vec{\mathcal{G}}_{A,n-1}$ (for $A = O, S, X, T$), the subsequent iteration of the guess is calculated by

$$\vec{\mathcal{G}}_{A,n} = [\mathrm{M}]_A \left( \vec{\mathcal{H}} * \Pi_{i \neq A} * \vec{\mathcal{G}}_{i,n-1} \right) \tag{5}$$

where $A = X, S, T$ and $O$, and $[\mathrm{M}]_A$ is the matrix that projects onto the subspace spanned by the codebook of the objects, location, time and size for $A = O, X, T$ and $S$ respectively. The process converges to the correct factorization for large $D$ if the hypervectors are sufficiently random. However, if the lengthscale $w_i$ of correlations is large, then the resonator network will converge to random results.

## 7 EXPERIMENTAL RESULTS

### 7.1 EXPERIMENTAL SETUP

We perform experiments for detailed exploration of how various settings in encoding influence the HDC performance for both learning and cognitive information retrieval. We select MNIST handwritten digits as our main and run all experiments in the following sections using the PyTorch framework on the Intel Core i7-12700K platform.

### 7.2 ENCODING: LEARNING VS. COGNITION

In this section, we discuss our expectations on how learning and decoding efficiency vary as a function of $w_i$. For a large value of $w_i$, the similarity of the position vector does not change over large distances. So the feature values are averaged out and would be better for learning. However, making $w_i$ too large would make the learning a full average of the feature which would contain too little information about the data to learn the differences efficiently. If we consider decoding, on the other hand, we would like the location ID vectors to be nearly orthogonal. As a result, the noise term in the decoding would be very low and would enable a perfect recovery of the data points. The main question thus is, what is the optimal value of $w_i$. To understand this, we can study the probability distribution of when the feature values are either $1$ or $0$, alongside the joint probability distributions of the feature locations. What is most important is how accurately we can decode where the feature values are $1$, since they contain the information about our features. So,

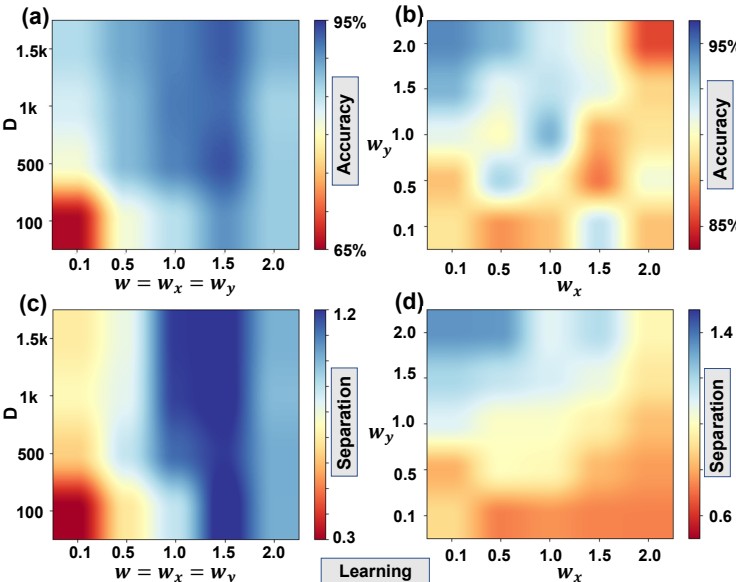

Figure 8: The learning accuracy as a (a) function of $D$ and $w$ (b) function of $w_x$ and $w_y$; the learning separation as a (c) function of $D$ and $w$ (d) function of $w_x$ and $w_y$

we need to study how the locations of 1 are correlated over positions in the image. To do this, we first consider two coordinates $(x, y)$ and $(X, Y)$. We first construct the probability function $p^1(x, X, Y) = \mathbb{P}(f_{x,Y} = 1 \& f_{X,Y} = 1)$ based on numerical data set. This is the probability that both the conditions $f_{x,Y} = 1$ and $f_{X,Y} = 1$ hold true. This function measures the correlation between two pixels that are separated horizontally at different $x-$location, but same vertical coordinate. Similarly, we construct $p^2(y, X, Y) = \mathbb{P}(f_{X,y} = 1 \& f_{X,Y} = 1)$, which measures the vertical correlations of the pixels. Together, these probability distributions show us how the features are correlated in a certain direction. Using this, we can calculate the average value of $l_x = \langle |x - X| \rangle$ and $l_y = \langle |y - Y| \rangle$. Thus, using this we can heuristically estimate that $w_x = l_x$ and $w_y = l_y$ would be the optimal choice for the scale of the kernels.

## 7.3 Cognition: HDC Decodability

We present the results of decoding the hypervectors back to feature space as a function of $w$ and dimension $D$ in Fig. 7(a). We chose $w_1 = w_2 = w$. The accuracy ranges from 85% at the lowest to 100% for the highest accuracy. We see that at $w = 1.5$ there is a boundary, and for $w > 0.75$ the decoding accuracy falls sharply. Moreover, for small dimensions, the accuracy decreases. We also show the accuracy as a function of $w_x$ and $w_y$, at a fixed dimension of $D = 500$ in Fig. 7(b). We see that $w_y$ has a boundary of 1.0 after which the accuracy falls, and $w_x$ has a boundary at 1.5. For small $w$, the decoding process would be very accurate since it can distinguish between nearby feature values independently. When $w$ increases, however, the decoding process cannot distinguish between nearby encoded feature values, resulting in inaccuracies. This is reflected in Fig. 7(a) where we see that there is a vertical line at $w \sim 1.5$ through which the accuracy drastically reduces.

We also show the decoding separation in Fig. 7(c) and (d) for the corresponding accuracy plots. When the separation metric is low, the accuracy is high. The correlation, however, loses meaning when the separation is large and the accuracy is high because at these levels the overlap will vary randomly for different data sets, which we observe at high accuracy ($> 85\%$) where the separation varies with no corresponding change in accuracy.

## 7.4 HDC Learnability

We present the result for the accuracy of learning of the classes from the quantized hypervectors. We show the heatmap for the accuracy of classification as a function of $w$ and dimension $D$ in Fig. 8(a). We chose $w_1 = w_2 = w$. For a large dimension where the learning model converges to the optimal capacity, there is a maximum for the accuracy at $w = 0.5$ We also observe the accuracy as a function

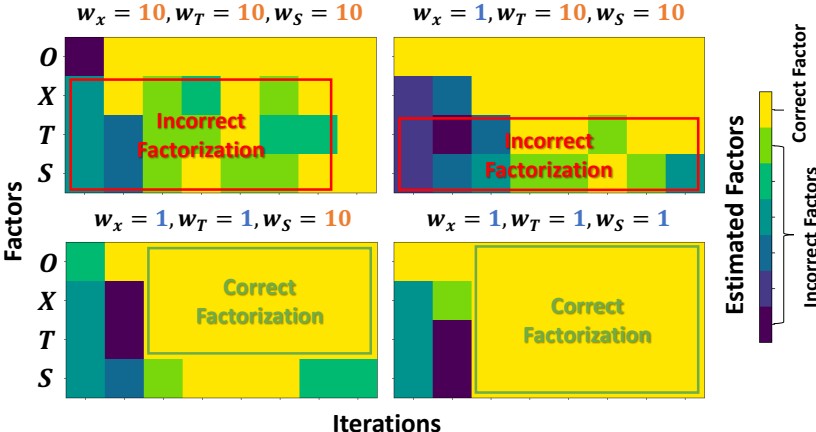

Figure 9: Solution to the resonator network with four factors, with each factor continuously encoded using the random feature encoding.

of $w_x$ and $w_y$ in Fig. 8(b), at fixed dimension. We find that a large $w_x$ and $w_y$ are preferred for good accuracy. However, the accuracy does not improve very fast by increasing $w_y$, unlike the case of $w_x$.

The trend for learning as a function of $w$ is expected based on theoretical arguments. We focus on $D = 1.5k$, which is large enough to avoid noise issues due to low dimensions. Here, for small $w$, the encoding maps every data point to an orthogonal vector in the hyperspace and thus takes up a lot of capacity. As a result, the correlations between the vectors in the same class will be quite low. In this case, the learning will be inefficient and inaccurate. Conversely, with large $w$, the encoding maps every data point to very correlated hypervectors in hyperspace. As a result, the learning process will not be able to distinguish between the vectors of different classes, which results in low accuracy. We also show the separation in Fig. 8(c) and (d) for the corresponding learning plots showing a good correlation at low accuracy and intermediary accuracy regions.

### 7.5 HDC FACTORIZATION PROBLEM

Finally, we discuss the effect of correlation on the HDC factorization problem using the resonator network, Sec. 6. The position, location and time values are continuously encoded, with the values chosen from the set $\{1, 2, ..., 10\}$, while the objects are represented by random vectors. We set the length scale $w_i = 10$ initially for the three continuous factors (so that the vectors representing various nearby values remain highly correlated), and then change the value to $w_i = 1$ sequentially.

Fig. 9 shows the solution of the resonator network as the function of iteration. Each color represents a specific index in the codebook for each factor, and the correct factorization is where all the factors are yellow. For the first experiment, we set all $w_i = 1$, resulting in the $S, T, X$ factors being highly correlative. Thus resonator network converges onto a random result for those factors since it cannot distinguish between hypervectors representing different values. Next, we set $w_X = 1$ so that the hypervectors representing different positions are uncorrelated, while the hypervectors representing time and size remain highly correlated. In this case, the object and position are correctly decoded, while the time and size are randomly decoded. In the last two cases, we set $w_T = 1$ and $w_S = 1$, respectively. The corresponding factors are also correctly decoded, highlighting the importance of controlling the correlation in HDC factorization.

## 8 CONCLUSION

This paper proposes a universal hyperdimensional encoding method that can be easily adapted to both learning and cognitive tasks. We provide an extensive exploration of how various settings in encoding influence the performance of HDC in both tasks. We highlight the distinction between learning high-level patterns and information retrieval from the angle of HDC operations. Our encoding can optimally separate or correlate encoded data points in high-dimensional space for downstream tasks.

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

## A  HYPERDIMENSIONAL COMPUTING: AN OVERVIEW

The brain's circuits are massive in numbers of neurons and synapses, suggesting that large circuits are fundamental to the brain's computing.

HDC Kanerva (2009) explores this idea by looking at computing with high-dimensional vector representations, or hypervectors. As the fundamental units of HDC, hypervectors are constructed

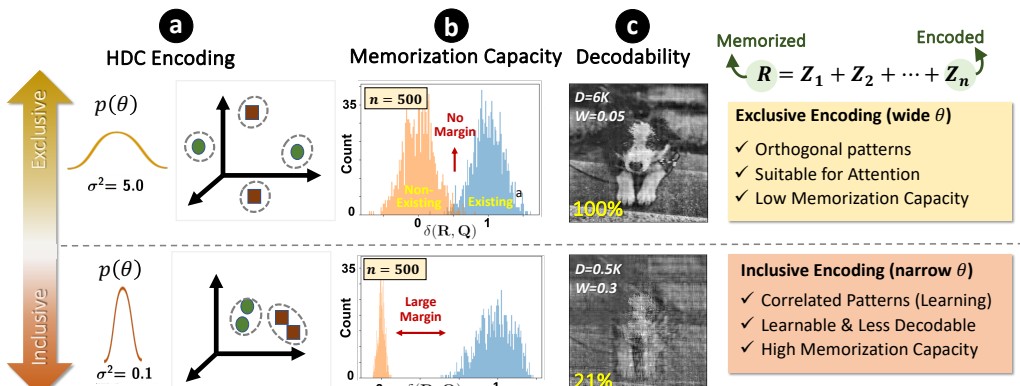

Figure 10: Two directions in HDC encoding designs: the correlative one is suitable for learning and the exclusive one is suitable for cognition.

from raw signals using an encoding procedure. Within a hyperspace, many different, nearly orthogonal hypervectors exist with dimensionality in the thousands Kanerva (1998). This lets us combine such hypervectors into a new hypervector using well-defined vector space operations, e.g., *Bundling (+)* and *Binding (∗)*. Bundling uses element-wise addition to represent sets, and binding expresses conjunctive association with element-wise multiplication. Hypervectors are holographic and (pseudo)random with i.i.d. components. More specifically, they combine and spread information across all its components in a full holistic representation so that no element is more responsible for storing any piece of information than another.

In recent years, HDC has been employed in a range of applications, such as classification Kanerva (2009), activity recognition Kim et al. (2018), biomedical signal processing Moin et al. (2021), multimodal sensor fusion Räsänen & Saarinen (2015), security Thapa et al. (2021); Zhang et al. (2021) and distributed sensors Kleyko et al. (2018). A key HDC advantage lies in the capability of training in one or few shots, where object categories are learned from a few examples without many iterations. HDC has achieved comparable or higher accuracy compared to support vector machines (SVMs), gradient boosting, and convolutional neural networks (CNNs) Rahimi et al. (2018); Mitrokhin et al. (2019), as well as lower execution energy on embedded processors compared to SVMs and CNNs Montagna et al. (2018).

In these successful HDC applications, HDC encoding is essential to the quality of computing. With a suitable encoding, information from inputs is well maintained to satisfy the needs of tasks. The encoding determines the following factors: (1) the distance metric for encoded data points and (2) the level of correlation or exclusiveness preserved after mapping to hyperspace. Despite the general success in HDC encoding, there are no guidelines on setting these factors in practice. In this work, we observe that a suitable HDC encoding design is application-specific. In Fig. 10, we categorize HDC applications into learning and cognition, then we shed light on two corresponding directions in HDC encoding.

- **Learning:** aims to capture the general pattern of data. It operates over encoded hypervectors, where the information of original data is preserved. Therefore, the usefulness of information maintained in hypervectors determines the learning quality. For learning, the HDC encoding needs to be inclusive and correlative. In other words, the encoding should abstract common information by keeping the similarity of neighboring data points in hyperspace, and we refer to this as the *Correlative Encoding*. This encoder correlates hypervectors under a certain distance metric, and similar data points are coarsely clustered in hyperspace. This clustering effect helps classify data that are not linearly separable and avoid overfitting with a smoother boundary. The inclusive encoder also theoretically increases the memorization and learning capacity of hypervectors (see Section 5). Generally, HDC encoding for learning does not require the exact decoding of information. Instead, it only needs a coarse differentiation between different patterns. Therefore, the learned models are often much more compact than the original training dataset.

- **Cognition:** aims to represent structural data using neural patterns, which accordingly enables brain-like analysis and information extraction. The primary task of encoding in cognitive tasks is to represent data points exclusively instead of inclusively in hyperspace. We name this kind

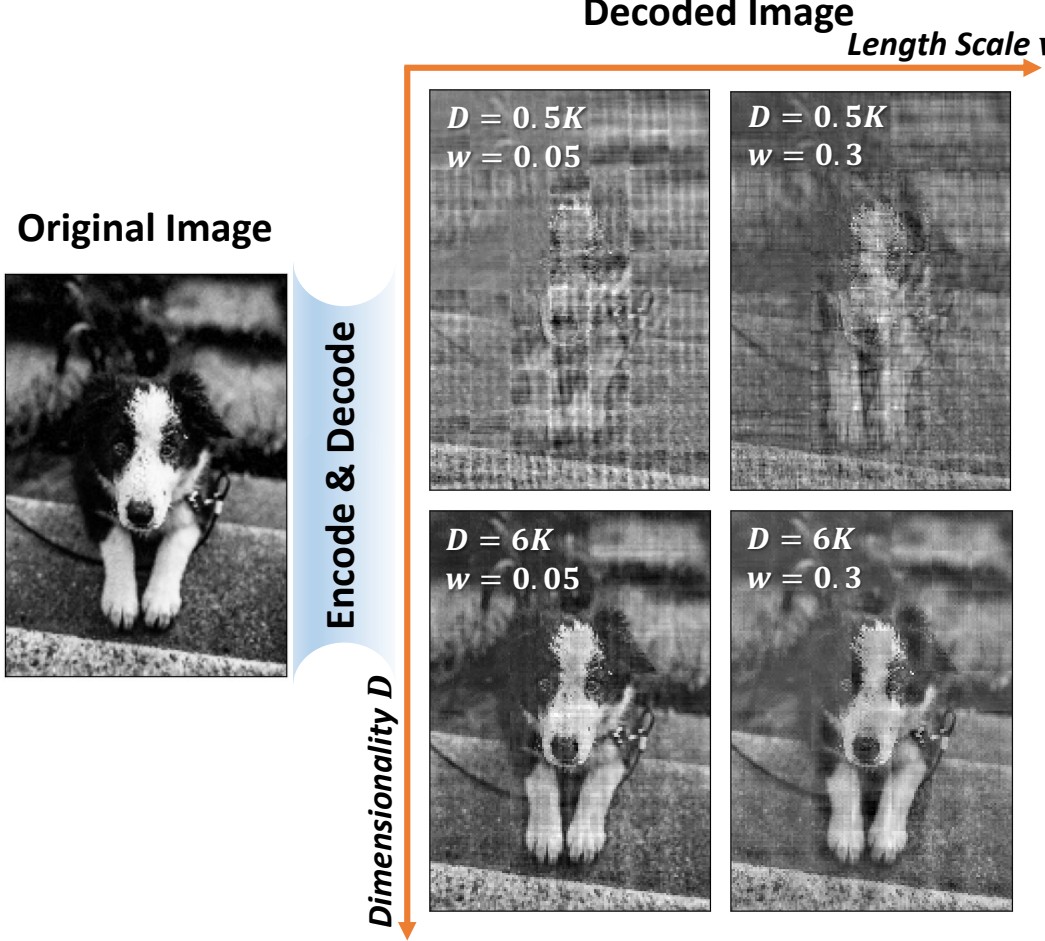

Figure 11: Example of the effect of dimensionality and width of the distribution on the decodability of an image.

of encoding as the *Exclusive Encoding*. This ensures accurate knowledge extraction such that memorized information can be used as prior information for various cognitive computation tasks. For example, prior research shows that exact memorization enables brain-like reasoning Poduval et al. (2022b) and perception Mitrokhin et al. (2019). To conclude, for cognitive operations, the key is the invertibility of hypervectors. The encoded information needs to be accurately decoded back to the original space to answer cognitive questions.

As shown in Fig. 10, learning and cognitive computation have different requirements for HDC encoders. For learning, the encoding is inclusive and correlative, preserving the similarity of nearby data. In contrast, encoding is exclusive for cognitive computation, where data points are mapped to orthogonal space and distinct. However, data decoding has completely different requirements as compared to learning. The encoding used for cognitive operation requires preserving enough information to ensure high decoding capability.

In Fig. 11, we present the image decoding results using our universal encoder with different tuning. We evaluate our proposed encoder with this practical image of a dog, then we try to accurately retrieve the original image from the encoded version. We do this however by separating the image into a few sub-images and performing the encoding-decoding procedure. We scan through different dimensionality and length scale settings, which determine the capacity and decodability of the encoded hypervector. Our results show that the decoding quality is significantly better when $D = 6k$ and $w = 0.05$, which is expected from our previous discussion. The larger the dimensionality a hypervector uses, the larger the decoding capacity it has. On the other hand, a small length scale is preferred if the capacity is large enough, such that it can maximize the separation. If the capacity is

not enough, then the noise from the separated location vectors will be significant and result in a high error noise.

## B  HDC WITH RANDOM VECTORS

In this section, we introduce the Hyperdimensional learning algorithm and provide insights on how to adjust the HDC encoder for learning tasks. Let us assume $p$ random data points in the hyperspace, $\{\vec{\mathcal{H}}_1, \vec{\mathcal{H}}_2, \cdots, \vec{\mathcal{H}}_p\}$. Due to the randomness in high-dimension, these hypervectors are nearly orthogonal, that is, the similarity $\delta\langle \vec{\mathcal{H}}_i, \vec{\mathcal{H}}_j \rangle \approx 0$, where $1 \leq i \neq j \leq p$. HDC bundling operation combines these hypervectors into a single memory hypervector or model hypervector: $\vec{\mathcal{M}} = \sum_{i=1}^p \vec{\mathcal{H}}_i$. Its capacity depends on two factors: (1) the dimensionality of hypervectors, and (2) the correlation between the encoded data points. The randomness is also reflected within the similarity metric, which follows a Gaussian distribution with $\mu = 0$ and a non-zero $\sigma = \frac{1}{\sqrt{2D}}$ based on dimensionality $D$. Increasing the dimensionality further orthogonalizes these hypervectors, in the sense that the distribution is more squeezed with $\sigma$ decreasing. As a result, when we check whether a new encoded query $\vec{\mathcal{Q}}$ belongs to the memory hypervector $\vec{\mathcal{M}}$, we calculate the similarity $\delta(\vec{\mathcal{Q}}, \vec{\mathcal{M}}) = \sum_{i=1}^p \delta(\vec{\mathcal{Q}}, \vec{\mathcal{H}}_i)$. This leads to a Gaussian distribution with mean values being either 0 or 1 depending on the query: (1) $\vec{\mathcal{Q}}$ appears in those $p$ data points, which means that it will correctly match one component in the memory hypervector (signal) and mismatch with the rest (noise). (2) $\vec{\mathcal{Q}}$ is not part of $p$ data points, and the similarity value is essentially the sum of $p$ random values sampled from the Normal distribution. Therefore, the resulting similarity follows the Gaussian distribution with $\mu = 1$ in the case (1) and $\mu = 0$ in the case (2), with $\sigma = \sqrt{\frac{p}{2D}}$ for both cases.

## C  SEPARATION FOR LEARNING VS DECODING

The separation is similar to Eq. 6, which also measured the difference between the signal and noise distribution.

$$s = \frac{\mu_2 - \mu_1}{\sigma_1 + \sigma_2}, \tag{6}$$

However, the context between the two different separation values are very different. The former separation metric defined the difference of the noise distribution and a specific component of the feature vector. However, this definition of the separation metric calculates the separation between the distribution of the similar class element with an element that belongs to the same class, and the distribution of the similarity of the class element with a element of the different class.

