# OpenReview forum: "Optimal Hyperdimensional Representation for Learning and Cognitive Computation"
_ICLR.cc/2025/Conference — Submitted to ICLR 2025_

### Official Review · Reviewer_axtq · 2024-10-17

**Soundness:** 2
**Presentation:** 2
**Contribution:** 2
**Rating:** 5
**Confidence:** 4

**Summary:**

This paper proposes a novel hyperdimensional encoder, which demonstrates its flexibility and accuracy in learning and cognitive tasks through theoretical analysis and experimental verification, aiming to improve the efficiency and quality of information encoding in different application scenarios.

**Strengths:**

The advantages of this paper are its high flexibility and cost-effectiveness, which can adapt to diverse application scenarios while supporting efficient learning and accurate information retrieval. In addition, it is based on a solid theoretical foundation and optimizes performance through parameter adjustment, ensuring effectiveness and accuracy in various tasks.

**Weaknesses:**

1. This article lacks some necessary details. Please provide specific experimental steps and details, such as various hyperparameters of the experiment, such as learning rate and training step size.
2. Can this article be experimented on a larger dataset such as cifar10? The current experimental results are only conducted on mnist, which is a bit too simple.

**Questions:**

see weakness bellow.

---

### Official Review · Reviewer_4Mst · 2024-11-01

**Soundness:** 2
**Presentation:** 3
**Contribution:** 2
**Rating:** 3
**Confidence:** 4

**Summary:**

The paper proposes a 'universal' encoding framework for hyperdimensional computing systems that claims to optimally balance the tradeoff between correlative learning and structural representational learning (cognition). This system makes use of the high-dimensional nature of hypervectors to generate orthogonal hypervectors in the encoding framework through Gaussian sampling in the encoding process. This results in class hypervectors that are mutually orthogonal and whose distributions are separated using the separation metric provided in the paper. Memorizing associations between hypervectors and triples of datapoints (or data vectors) is conducted using a resonator network. whose solutions are the classification outputs of the HD system.

The paper then presents data for the MNIST image classification dataset, aiming to use the given hyperdimensional encoding scheme and resonator network decoder. Image classification accuracy is seen to sharply rise for this test case. The cognition/learning tradeoff is likewise explored in the ablation studies, with image classification decoder accuracy therefore ranging from 85-100%.

**Strengths:**

The strengths of the paper are as follows:
1) It is highly original in my eyes in that it establishes the learning/cognition tradeoff for hyperdimensional systems (I have not seen prior work do this). This tradeoff is then put into theoretically grounded form and used to build a separation metric, a orthogonal class hypervector set and an encoder that balances correlative learning using a Gaussian sampling metric with learning structural features.
2) The notation is clear and concise, and the methods are clearly explained. Presentation is therefore while not exceptional - I would prefer more detail on cognition/learning tradeoffs and a conceptual example - solidly meets the bar for this conference.
3) It is significant in that it strongly establishes (1), but other than that I find the experimental results to be concerning/lacking. I will elaborate in the Weaknesses section. I believe it is a significant work but would strongly benefit from further validation.

**Weaknesses:**

1) The paper claims to present a universal encoding, but applies this solely to image data. This is not just an experimental section issue - the Methods section repeatedly references cross-pixel correlation, for instance. I would advise that the paper's claims be walked back to image classification or image encodings, considering that recent work in hyperdimensional computing has examined sensor data streams in Kalman filtering [1], time series data [2] and more recently, symbolic learning [3]. Either that or ablation studies on the generalizability of this encoding framework to non-image data or symbolic learning tasks would be appreciated.
[1] https://ieeexplore.ieee.org/document/10473878
[2] https://arxiv.org/abs/2402.01999
[3] https://www.nature.com/articles/s42256-023-00630-8

2) The paper does present a theoretical case quite well. However, the paper validates its scheme on MNIST, which is commonly considered a prototyping dataset due to its easy separability between classes and its wide class boundaries. Datasets such as CIFAR-10 have much more nonlinear class boundaries and I would be very interested to see the performance of this system for such datasets. Especially since recent work [4] has shown ~95% accuracy on MNIST and also validated on CIFAR10. I would ask what the significance in increased performance (i.e., validation benefits) of this system are?
[4] https://arxiv.org/pdf/2203.09680

3) The proposed cognition/learning tradeoff does not quite address the claim of 'optimal' encoding - I do not see a proof of the \textit{optimality} of the encoding formulation, merely a very elegant proof of the two axes along which HD learning proceeds. Before accepting a paper with broad claims to optimality of encoding formulations I would want further proof on a more general set of data assumptions (rather than merely image data under the assumptions provided).

**Questions:**

1) The paper claims to present a universal encoding, but applies this solely to image data. This is not just an experimental section issue - the Methods section repeatedly references cross-pixel correlation, for instance. I would advise that the paper's claims be walked back to image classification or image encodings, considering that recent work in hyperdimensional computing has examined sensor data streams in Kalman filtering [1], time series data [2] and more recently, symbolic learning [3]. Either that or ablation studies on the generalizability of this encoding framework to non-image data or symbolic learning tasks would be appreciated.
[1] https://ieeexplore.ieee.org/document/10473878
[2] https://arxiv.org/abs/2402.01999
[3] https://www.nature.com/articles/s42256-023-00630-8

2) The paper does present a theoretical case quite well. However, the paper validates its scheme on MNIST, which is commonly considered a prototyping dataset due to its easy separability between classes and its wide class boundaries. Datasets such as CIFAR-10 have much more nonlinear class boundaries and I would be very interested to see the performance of this system for such datasets. Especially since recent work [4] has shown ~95% accuracy on MNIST and also validated on CIFAR10. I would ask what the significance in increased performance (i.e., validation benefits) of this system are?
[4] https://arxiv.org/pdf/2203.09680

3) For data that is autoregressive, this system may in fact harm accuracy if the noise added from the Gaussian kernel sampling during encoding is excessive. This would affect the encoding's applicability to time series data. Can the authors comment on this?

4) Can the authors comment on a theoretically grounded value of w? What I can see is that w appears to be dataset centric or data driven, and would therefore harm the claims to an optimal, universal encoding formulation.

---

### Official Review · Reviewer_NeiD · 2024-11-04

**Soundness:** 1
**Presentation:** 2
**Contribution:** 1
**Rating:** 1
**Confidence:** 5

**Summary:**

This paper aims to improve hyperdimensional computing (HDC) encoding based on kernels. They presented empirical results using a small synthetic image classification dataset.

**Strengths:**

The paper explored HDC encoders based on kernel methods for synthetic image classification and recall tasks. However, the use of kernel methods in HDC has already been investigated in detail (see weakness).

**Weaknesses:**

1) The use of kernel methods to enhance encoding has been investigated in HDC since 2021 hence it is not a novelty. Here are some pointers, especially Vector Function Architectures provides an in-depth theoretical analysis of the initialization and its impact on the shape of kernels.

- [Vector Function Architectures] E. Paxon Frady and Denis Kleyko and Christopher J. Kymn and Bruno A. Olshausen and Friedrich T. Sommer, "Computing on Functions Using Randomized Vector Representations", arXiv:2109.03429

- E Paxon Frady, Denis Kleyko, Christopher J Kymn, Bruno A Olshausen, Friedrich T Sommer, "Computing on functions using randomized vector representations (in brief)", Annual Neuro-Inspired Computational Elements Conference, 2022.

- [Spatial Semantic Pointers] Furlong, P.M., Eliasmith, C. Modelling neural probabilistic computation using vector symbolic architectures. Cogn Neurodyn (2023)

2) The presented results are very weak using only synthetic image classification datasets, and binary classes, reaching 95%! There is no report with other established datasets (CIFAR, Imagenet, etc) and it also lacks pointing to strong baselines.

3) There are several misreferences. Here is a list:

- Holographic Reduced Representations (Tay et al., 2019) --> Plate, T. A. (1995). Holographic Reduced Representations. IEEE Transactions on Neural Networks, 6(3):623–641

- Multiply-Add-Permute (Kleyko et al., 2021) --> Gayler, R. W. (1998). Multiplicative Binding, Representation Operators & Analogy. In Advances in Analogy Research: Integration of Theory and Data from the Cognitive, Computational, and Neural Sciences, pages 1–4.

- Binary Spatter Codes (Kleyko et al., 2016) --> Kanerva, P. (1994). The Spatter Code for Encoding Concepts at Many Levels. In International Conference on Artificial Neural Networks (ICANN), pages 226–229.

**Questions:**

See weakness

---

### Meta-Review · Area_Chair_zkhR · 2024-12-18

**Metareview:**

The three reviewers generally agree that the paper does not meet the acceptance standards of the venue. While one reviewer offers a marginally below-threshold rating, the overall consensus is negative. The main criticisms concern the lack of novelty relative to prior work in hyperdimensional computing (HDC) with kernel-based encodings, insufficient experimental validation on established and more challenging datasets, limited contextualization against known baselines, and a lack of clarity and detail in certain methodological and experimental aspects.

**Additional Comments On Reviewer Discussion:**

The authors did not post any rebuttal. There was no discussion.

---

### Decision · Program_Chairs · 2025-01-22

Reject